# Characterization of a Novel Mitovirus of the Sand Fly *Lutzomyia longipalpis* Using Genomic and Virus–Host Interaction Signatures

**DOI:** 10.3390/v13010009

**Published:** 2020-12-23

**Authors:** Paula Fonseca, Flavia Ferreira, Felipe da Silva, Liliane Santana Oliveira, João Trindade Marques, Aristóteles Goes-Neto, Eric Aguiar, Arthur Gruber

**Affiliations:** 1Department of Microbiology, Instituto de Ciências Biológicas, Universidade Federal de Minas Gerais, Belo Horizonte 30270-901, Brazil; camargos.paulaluize@gmail.com (P.F.); arigoesneto@gmail.com (A.G-N.); 2Department of Biochemistry and Immunology, Instituto de Ciências Biológicas, Universidade Federal de Minas Gerais, Belo Horizonte 30270-901, Brazil; fvianaferreira@gmail.com (F.F.); jtm@ufmg.br (J.T.M.); 3Bioinformatics Postgraduate Program, Instituto de Ciências Biológicas, Universidade Federal de Minas Gerais, Belo Horizonte 30270-901, Brazil; felselva@gmail.com; 4Bioinformatics Postgraduate Program, Universidade de São Paulo, São Paulo 05508-000, Brazil; liliane.sntn@gmail.com; 5Department of Parasitology, Instituto de Ciências Biomédicas, Universidade de São Paulo, São Paulo 05508-000, Brazil; 6CNRS UPR9022, Inserm U1257, Université de Strasbourg, 67084 Strasbourg, France; 7Department of Biological Science (DCB), Center of Biotechnology and Genetics (CBG), State University of Santa Cruz (UESC), Rodovia Ilhéus-Itabuna km 16, Ilhéus 45652-900, Brazil; 8European Virus Bioinformatics Center, Leutragraben 1, 07743 Jena, Germany

**Keywords:** *Mitovirus*, positive-sense single-stranded RNA viruses, mitochondrial viruses, RNA interference, nucleotide frequency, codon usage, seed-driven progressive assembly, profile Hidden Markov Model

## Abstract

Hematophagous insects act as the major reservoirs of infectious agents due to their intimate contact with a large variety of vertebrate hosts. *Lutzomyia longipalpis* is the main vector of *Leishmania chagasi* in the New World, but its role as a host of viruses is poorly understood. In this work, *Lu. longipalpis* RNA libraries were subjected to progressive assembly using viral profile HMMs as seeds. A sequence phylogenetically related to fungal viruses of the genus *Mitovirus* was identified and this novel virus was named Lul-MV-1. The 2697-base genome presents a single gene coding for an RNA-directed RNA polymerase with an organellar genetic code. To determine the possible host of Lul-MV-1, we analyzed the molecular characteristics of the viral genome. Dinucleotide composition and codon usage showed profiles similar to mitochondrial DNA of invertebrate hosts. Also, the virus-derived small RNA profile was consistent with the activation of the siRNA pathway, with size distribution and 5′ base enrichment analogous to those observed in viruses of sand flies, reinforcing *Lu. longipalpis* as a putative host. Finally, RT-PCR of different insect pools and sequences of public *Lu. longipalpis* RNA libraries confirmed the high prevalence of Lul-MV-1. This is the first report of a mitovirus infecting an insect host.

## 1. Introduction

Viruses are the most abundant biological entities in the biosphere, being found in every environment and infecting a wide range of organisms, such as plants, insects, mammals, and microorganisms [1,2,3]. Surveys to detect, identify, and characterize viral diversity are challenging due to the limited ability to isolate and grow viruses and their hosts in laboratory [4]. Furthermore, viruses do not have universally conserved sequences in their genomes that can be used as targets for PCR-based assays, such as the ribosomal genes of prokaryotes and eukaryotes [5,6]. Finally, viruses present much higher evolutionary rates than prokaryotes and eukaryotes [7,8,9,10], which often implies that novel viruses are just too divergent to be detected by serological and molecular assays designed for specific known pathogens [6,11]. Metagenomics was classically defined as a sequence analysis method using samples containing multiple organisms [12]. With the advent of high-throughput sequencing platforms, metagenomics has greatly accelerated the pace of genome characterization and detection of viruses and hosts from environmental and clinical samples, without the need for isolation and prior cultivation [6,11,13,14]. Such an approach has allowed researchers to unveil viral diversity and virus–host interactions in many eukaryotic and prokaryotic organisms [15,16,17].

The relationship between viruses and their hosts is considered a coevolutionary process since viruses are obligate intracellular parasites and require the host’s cellular machinery for protein synthesis. Also, viruses are subject to the same evolutionary pressures that shape the host genome composition and codon usage [18,19]. In fact, the coding regions of hosts and viruses tend to share common compositional features, such as dinucleotide composition and codon usage patterns [18,19,20]. Dinucleotide under- and over-representations are among the most studied and relevant of these patterns, and can be used to infer ecological functions and to classify viruses [21,22]. Nevertheless, dinucleotide composition can eventually be more related to the characteristics of the virus family than to the specific viral host species [23].

Hosts and their viruses are under constant adaptation and selective pressure, which is led by hosts developing new defense strategies and viruses developing new infection and counter-defense strategies [24]. One of the strategies developed by eukaryotic organisms against viral infections relies on the RNA interference (RNAi) pathways. RNAi pathways are mechanisms that induce silencing of self and non-self RNAs based on sequence-specific homology using small RNAs (sRNAs) [25]. In insects, it is well known the existence of three separate RNAi pathways- micro-RNA (miRNA), piwi-interacting RNA (piRNA), and small interference RNA (siRNA), with the latter being described as a hallmark of antiviral response in these organisms [26,27]. The siRNAs pathway is activated when double-stranded RNA is recognized and processed by the enzyme Dicer-2 into 19–23 nt long duplex of sRNAs. These sRNAs, are then loaded into Argonaute-2 to generate the small interferent RNA-induced silencing complex (siRISC) that produces virus-derived small RNAs, which in turn are used to find and cleave complementary RNAs. Interestingly, our group has shown that the size profiles of virus derived small RNAs are unique and distinctive, depending on the combination of host and virus species. Such feature can be used to determine the origin of viral sequences identified in metagenomic samples [25,27] and to differentiate between endogenous and exogenous viral sequences, which is a major problem in studies based solely on long RNA sequencing [25,26,28].

Recent studies have revealed that insects exhibit an extraordinary diversity and abundance of viruses [15]. Among invertebrates, insect vectors such as mosquitoes and phlebotomies have been extensively studied since they are associated with the transmission of several viral pathogens that threaten human health [29,30]. Sandflies are insects belonging to the order Diptera, subfamily Phlebotominae, which present hematophagous feeding habits. There are circa 900 species already described, and 70 of these species have been reported as potential vectors of *Leishmania* spp. with few others involved in the natural transmission of viruses, such as *Phlebovirus* (*Reoviridae* family) [31]. These insects are still capable of harboring other microorganisms, since they have contact with different environments and substrates [32,33,34,35]. This aspect can be especially relevant since many of these microorganisms that make up insect microbiota can also carry viruses.

Viruses of the genus *Mitovirus* were formerly classified [36] in the family *Narnaviridae*, together with the genus *Narnavirus* [37]. Both genera show distinct subcellular localizations, comprise capsidless viruses with a monopartite positive sense single-stranded RNA genome of 2.3–2.9 kb, and present a single gene encoding an RNA-dependent RNA polymerase (RdRp). In the case of *Mitovirus*, the RdRp gene presents a mitochondrial-type codon usage, with UGA coding for tryptophan. In the current International Committee on Taxonomy of Viruses (ICTV) report, as of March of 2020—Master Species List #35 [38], *Narnaviridae* family, containing the genus *Narnavirus*, was included in the *Wolframvirales* order, whereas genus *Mitovirus* now belongs to a newly created *Mitoviridae* family, order *Crippavirales*. Both orders are currently members of the *Lenarviricota* phylum, which also comprises *Leviviridae* (order *Levivirales*), a family of positive-sense single-stranded RNA bacteriophages that may have originated narnaviruses, mitoviruses, and ourmiaviruses [39]. Mitoviruses have been identified in many fungal hosts, such as *Entomophthora muscae* and *Fusarium boothii* [40,41] and narnaviruses in invertebrates such as insects and other arthropods [15], but, differently from mitoviruses, their replication occurs in the cytoplasm of the fungal hosts [37,42,43]. In addition to *Narnaviridae* family, fungi can also be infected with other viruses (mycoviruses), which replicate in the cytoplasm of the host cells [44]. Similar to *Narnaviriridae* and *Mitoviridae*, the *Botourmiaviridae* family is composed of viruses infecting the cytoplasm of plant cells and, unlike these two families, their genome is composed of three monocistronic segments [45]. Nevertheless, some studies have identified members of the *Ourmiaviridae* family infecting the cytoplasm of filamentous fungi [46].

Although there is an increasing number of reports uncovering the diversity of viruses circulating in insects, they are mainly restricted to nucleic acid sequencing, with no additional biological characterization, thus restricting the ability to determine the origin of the viral sequences [2,15]. Also, most of the viral surveys reported in the literature rely on the use of conventional pairwise similarity searches, which often yield no identification of the sequences found [47]. It has been demonstrated that pairwise similarity searches are effective in detecting relatively close homologs, but fail to identify distantly related sequences [48]. Conversely, similarity methods using sequence profiles are able to detect remote homologs with much higher sensitivity [49]. Profile Hidden Markov Models (profile HMMs) are probabilistic models built from multiple sequence alignments that cover the variability of residues in all positions, including indels and inserts [6]. Such models have been increasingly used in viral classification and discovery [50,51,52,53].

An additional challenge to detect novel viruses from metagenomics samples is to assemble large metagenomic datasets composed of an unknown number of different organisms. An alternative method for DNA assembly was described by our group and implemented on GenSeed-HMM [54], a program that uses profile HMMs as seeds for targeted progressive assembly. Such approach can be used in many applications, including viral discovery [6,54].

In this work, we investigate the diversity of viruses circulating in the sandfly *Lutzomyia longipalpis*, the most important vector of *Leishmania chagasi* in the New World. For this goal, we use three innovative methods: (1) profile HMMs to interrogate public long RNA sequencing data, (2) progressive assembly using profile HMMs as seeds, and (3) small RNA profiles to differentiate exogenous from endogenous viral sequences. Using this integrated approach, we identify and describe Lul-MV-1 (*Lutzomyia longipalpis mitovirus 1*), the first mitovirus found to infect the mitochondria of an insect host.

## 2. Materials and Methods

### 2.1. Acquisition and Processing of RNA Libraries

*Lu. longipalpis* public libraries of long and small RNAs were downloaded from the NCBI Sequence Read Archive (SRA) repository (https://www.ncbi.nlm.nih.gov/sra). Accession numbers of the analyzed libraries are listed in Appendix A. In total, six small RNA (sRNA) libraries and two long RNA (lRNA) libraries were used in this study. Libraries were submitted to quality end-trimming and adapter removal. Sequences presenting < 80% of bases with Phred quality below 20 or a length shorter than 20 bases were discarded. The remaining reads were used in further analyses.

### 2.2. Profile HMM Screening and Progressive Assembly

Long RNA reads of *Lu. longipalpis* public libraries (Appendix A) were used for viral sequence detection and reconstruction. A subset of 506 profile HMMs was manually built from the vFAM database [50]. These models were chosen based on their unequivocal functional annotation and for representing virus-specific proteins or sequences distantly related to prokaryotic or eukaryotic orthologs. We used HMM-Prospector (https://github.com/gruberlab/hmmprospector [accessed on 20 December 2020]), a Perl program to screen the profile HMMs against 6-frame translated versions of the long RNA datasets. HMM-Prospector uses hmmsearch program from HMMER package v. 3.1 [55] to run similarity searches and then processes the results, generating tabular files with qualitative and quantitative results. Profile HMMs detecting the highest numbers of significant hits (score > 30 and/or e-value < 1 × 10^−5^) were used as seeds for GenSeed-HMM [54], a tool for seed-driven progressive assembly. The reconstructed sequences were submitted to sequence similarity searches using BLASTX [56] against the non-redundant (nr) NCBI database. The programs Artemis (v. 16.0) [57] and InterProScan (version 5.36–75.0) [58] were used to detect open reading frames (ORFs) and conserved domains, respectively. Hits with e-values smaller than 1 × 10^−5^ for nucleotide comparison or 1 × 10^−3^ for protein comparison were considered significant. Viral genomic segments were classified as described [59].

### 2.3. Phylogenetic Analysis

A dataset composed of public protein sequences (Appendix A) related to mitoviruses, narnaviruses, and ourmiaviruses/ourmia-like viruses was constructed and submitted to a multiple sequence alignment with MUSCLE [60]. Phylogenetic reconstruction was performed by using IQ-TREE version 1.6.11 [61] with ModelFinder [62] to determine the model that minimizes the BIC (Bayesian Information Criterion) score. Node support values were determined using 1000 pseudoreplicates with the ultrafast bootstrap approximation (UFBoot) method [63]. The obtained trees were visualized and edited with Dendroscope [64].

### 2.4. Analysis of Small RNA Libraries

The pre-processing of sRNA libraries was performed as described [25]. Briefly, small RNAs reads were mapped against assembled contigs or viral genomes using Bowtie [65] allowing one mismatch. Small RNA size profile was calculated as the frequency of each small RNA read size mapped on the reference genome or contig sequence considering each polarity separately. We used a Z-score to normalize the small RNA size profile and to plot heatmaps for each sequence using R language (version 3.0.3) with gplots package (version 2.16.0). Pearson correlation with a confidence interval >95% of the Z-score values were computed to evaluate the relationships between the small RNA profiles from different contigs or reference genomes. The profile similarity was assessed using hierarchical clustering with UPGMA as the linkage criterion. Groups of sequences with more than one element with at least 0.8 of Pearson correlation between each other were assigned to clusters. Small RNA size profile, 5′ base enrichment, density of coverage and additional data analysis were evaluated using in-house Python, Perl, and R scripts. Statistics of 5′ base enrichment was calculated as described [66]. Similarities between small RNA size distributions were defined using hierarchical clustering with *K-means* as the linkage criterion in R using corrplot package [67]. Empirical cumulative frequency of small RNA size distribution was computed and compared using *ecdf* function built-in R software where the Kolmogorov–Smirnov test was used to determine statistical significance.

### 2.5. Dinucleotide and Codon Usage Analyses

Dinucleotide frequencies and codon usage were calculated using programs from the EMBOSS package (version 6.6.0) [68]. First, we used the program extractfeat to extract the coding sequences (CDS) from GenBank files and to store the data in FASTA format. Next, we used compseq to calculate the composition of unique 2-mer words in all frames to determine the dinucleotide expected/observed frequencies. Finally, the cusp program was used to generate codon usage tables containing the number of codons per 1000 bases, given the input sequence and the proportion of usage of each codon among its redundant set. The correlation between virus and host frequencies was calculated using the Pearson correlation test. Dinucleotide frequency was plotted using the R package corrplot, which grouped elements into clusters based on the results of the Pearson correlation test with a threshold above 0.8. Codon usage values were plotted as a heatmap with groups containing elements with mutual Pearson’s correlation coefficients of at least 0.8. The viral and mitochondrial genomes of fungal and insect hosts analyzed in this study are listed in Appendix A.

### 2.6. Amplification and Sanger Sequencing

To confirm the presence of the virus found in *Lu. longipalpis*, we used eight pools containing 10 sandflies per pool, collected from a colony originally started from individuals collected in Teresina, Brazil, and maintained at the Laboratory of Physiology of Hematophagous Insects (Department of Parasitology, Instituto de Ciências Biológicas, Universidade Federal de Minas Gerais). Total RNA was extracted from pools of 5 insects each using Trizol reagent according to the manufacturer’s protocol (Invitrogen—Thermo Fisher Scientific Inc., Waltham, MA, USA). Total RNA (1 μg) was reverse transcribed using 250 ng of random primers specific primers per reaction. The resulting cDNA was used as template for PCR reaction containing primers designed to amplify a product of 509 bp. Primer sequences are listed in Appendix A. Conventional PCR was performed using 1.5 μL of each designed primer (10 pmol/μL), 200 ng of cDNA or DNA and *Taq* DNA polymerase (Invitrogen, Thermo Fisher Scientific Inc.). PCR products were cleaned up with EDTA 125 mM precipitation protocol and sequenced using Sanger technology.

### 2.7. Analyses of Public Libraries

DNA and long and small RNA libraries were obtained from the NCBI SRA repository, listed with accession numbers in Appendix A. To estimate and analyze the abundance of the putative viruses, each library was compared to the viral and mitochondrial genomes using Bowtie2 [65]. The result was normalized by Reads Per Million (RPM) and plotted on a bar graph using the R program with the ggplot2 package [69].

## 3. Results

### 3.1. Identification of Viral Sequences in Lu. longipalpis Datasets

In a first attempt to detect possible viral sequences in *Lu. longipalpis*, we tested a set of 506 profile HMMs selected from the vFam database against two sequencing datasets of lRNA data, totaling 48 million reads. This strategy allowed us to select 18 different profile HMMs (Appendix A), which were used for progressive assembly, leading to the identification of 288 putative viral sequences. We observed a high abundance of contigs related to reverse transcriptase and integrase, in agreement with the high number of transposable and retroviral elements found in insect genomes [70] (Appendix A). In addition, we identified 18 contigs (vFam 561 and vFam 1529) derived from nucleoprotein N gene from *Rhabdovirus* (Appendix A), commonly found integrated in the genomes of many eukaryotes [71,72]. The other vFam models represented less than 10.1% of the identified contigs. Since the presence of symmetrical small RNAs of 20–23 nt, derived from viral sequences, is an indicative of activation of the siRNA pathway during viral replication [25], we decided to investigate the small RNA size distribution of all contigs reconstructed by profile HMM-seeded progressive assembly. From 288 assembled contigs representing putative viral sequences, only one sequence, reconstructed from vFam_571 model (Appendix A), presented a size distribution of small RNAs consistent with the activation of the siRNA pathway— symmetrical peak at position 21, derived from both strands without 5′ base preference (Figure 1).

The viral sequence assembled from vFam_571 presented a total length of 2697 nt with a GC content of 30.26%, corresponding to a monopartite ssRNA(+) genome. BLASTN searches against public databases showed no significant similarity. Using the universal genetic code, only short open reading frames (ORFs) were observed. BLASTX searches against the nr database revealed some similarity to RdRp of narna-like viruses such as the Wenling narna-like virus 9 (accession code YP_009337200) [15], which use UGA to encode tryptophan instead of signaling translation termination. In fact, when we switched to a mitochondrial genetic code, we found a long CDS coding for an 804-aa protein. InterProScan search of the protein sequence confirmed a positive identification for a mitoviral RdRp (InterPro entry IPR008686) and the presence of the Pfam domain PF05919. This nucleotide sequence corresponds to an almost complete genome sequence of a novel virus, hereinafter referred to as Lul-MV-1, the *Lutzomyia longipalpis* mitovirus 1.

### 3.2. Phylogenetic and Genome-Based Characterization of Lu. longipalpis Mitovirus 1

We performed a phylogenetic analysis with representatives of the current families *Mitoviridae* and *Narnaviridae* [38], who were formerly part of a common family [73], including some viral prototypic species ratified by the ICTV. We also included members of the genus *Ourmiavirus* (*Botourmiaviridae* family) and some unclassified viruses. Finally, two enterobacteria phages of the *Leviviridae* family were used as an outgroup, since these viruses are close relatives to *Mitovirus* and *Narnavirus* [74]. In addition, it has been proposed that these genera might have been evolved from *Leviviridae*-ancestors that infected bacterial endosymbionts, some of which may have generated mitochondria [39,45,75]. A phylogenetic reconstruction using RdRp sequences (Figure 2) revealed that Lul-MV-1 is closely related to the *Plasmopara viticola lesion-associated mitovirus 56* (QIR30279), a fungal mitovirus found in grapevine [76], and to a slightly lesser extent to the *Wenling narna-like virus 9* (YP_009337200), identified in crustaceans [15]. The narnaviruses constituted a sister clade composed of viruses infecting either fungal or insect hosts. The monophyly of the genera *Mitovirus* and *Narnavirus* was clearly supported by our analysis. Some viruses such as the *Grapevine associated narnavirus 1* (accession code CEZ26304), *Wenling narna-like virus 9* (YP_009337200), and the *Shahe narna-like virus 6* (APG77166), originally classified as narnaviruses, are members of the genus *Mitovirus* according to our analysis (Figure 2). Finally, members of the genus *Ourmiavirus* and some unclassified viruses constituted a sister clade to narnaviruses. In addition to the prototypical plant viruses such as the Cassava virus C (YP_003104770), *Ourmia melon virus* (YP_002019757) and *Epirus cherry virus* (YP_002019754), some recently described narna-like viruses are in fact ourmia-like viruses. Similarly, the *Aspergillus fumigatus mitovirus 1* (AXE72932), originally classified as a mitovirus [77], is clearly misclassified, and also belongs to the *Ourmiavirus*/ourmia-like clade. This group presents a high divergence among their members and it is possible that larger taxa samplings may indicate in the future that it is polyphyletic indeed.

### 3.3. Comparative Analysis of Structural and Compositional Features

In addition to the phylogenetic reconstruction, we performed a comparative analysis of the *Mitovirus* and *Narnavirus* genera according to the size of the viral genome and protein sequences (Figure 3). The coding sequence of the genus *Mitovirus* presents an average size of 2000–2500 nt and an ORF (Open Reading Frame) coding for an RdRp with a maximum size of 900 aa residues. Conversely, the *Narnavirus* genus shows an average genome size of 2500–3000 nt and an RdRp ORF with a maximum length of up to 1200 aa residues. Some narnaviruses also present ambigrammatic sequences, characterized by an additional large ORF coded in the reverse strand of the genome [78,79].

To characterize and propose a probable host of Lul-MV-1, we analyzed some intrinsic features of the viral genome. First, we compared the dinucleotide frequencies of Lul-MV-1 with those of mitochondrial genomes of fungi, mosquitoes (Culicidae) and phlebotomines (Phlebotominae). Lul-MV-1 presents a dinucleotide usage profile similar to the profiles observed in insects, but not to those of fungi (Figure 4). In fact, Lul-MV-1 presents a highly biased composition of the dinucleotides CC and GG and a mid-low GC bias, resembling the frequencies of insect mitochondria. A comparative clustering analysis of dinucleotide frequencies grouped Lul-MV-1 together with mitoviruses that infect fungi, whereas narnaviruses of fungi and insects were clearly grouped in another cluster (Appendix A). This result corroborates the genus assignment observed for Lul-MV-1 in the phylogenetic analysis (Figure 2). Finally, mitochondrial genomes of eukaryotic hosts were clustered into two closely related groups comprising fungi and insects, respectively (Appendix A).

Since di- and trinucleotides patterns can shape codon usage frequency in viruses [80], we also compared the codon usage profile of Lul-MV-1 with the profiles of some mitoviruses and mitochondrial genomes of fungi and insects, which in turn are the known or putative hosts of some viruses. Using a hierarchical clustering of these codon usage profiles, organisms presenting Pearson correlation values of at least 0.8 were grouped into the same clusters. From the six observed groups, clusters 1–3 are exclusively composed of mitoviruses infecting fungal hosts (Figure 5). Clusters 4 and 5 are composed of mitochondrial genomes of insects and fungi, respectively, with exception of Lul-MV-1, which is present in cluster 4, closely related to the mitochondrial genome of *Lu. longipalpis*. This result suggests that Lul-MV-1 is highly adapted to the mitochondrial codon usage of its putative host. The *Shahe narna-like virus 6* [15], characterized from a metagenomic dataset of crustaceans, showed a codon usage frequency closely related to the mitochondrial genome of the fungus *Plasmopara viticola*. Nevertheless, the *Plasmopara viticola associated mitovirus 56* presented a profile distantly related to its host, but closely related to other fungal mitoviruses. Finally, the *Wenling narna-like virus 7*, another virus detected in a metagenomic sample derived from crustaceans [15], showed a codon usage frequency unrelated to any of the tested organisms.

To better understand these results, we decided to restrict and deepen the analysis to UGA/UGG codons. Mitochondrial genomes often use a genetic code with UGA coding for tryptophan (Trp) rather than acting as stop codon. Thus, we chose a set of viral and mitochondrial genomes and determined the absolute counts and relative usage frequencies of these codons (Appendix A). In all cases where only one single UAG codon was counted, manual curation revealed that it was in fact acting as a stop codon at the end of coding sequence, rather than coding for tryptophan. In the case of the sampled narnaviruses, we did not observe the use of UGA(Trp) codons, either in fungal or insect viruses. More interestingly, this pattern was also seen for the hypothetical genes present in the reverse frame of some of these viral genomes. Conversely, the mitochondrial genomes of the corresponding fungal and insect hosts used almost exclusively UGA(Trp). This discrepant codon usage was highly correlated with the AT (adenine-thymine) content, with hosts’ mitochondria showing very high AT content, in the range of 70–80%, whereas the viruses presented much lower values, around 40%.

Mitoviruses, which are known to locate exclusively in the mitochondria of their hosts, showed a variable ratio of UGA/UGG codons (Appendix A), with a relatively close correlation with the ratios observed in the respective host’s mitochondrion. For instance, *Cryphonectria mitovirus 1* and *Cryphonectria parasitica* mitochondrion used UGA in 52.9 and 82.1% of the tryptophan codons. In another case, *Sclerotinia sclerotiorum mitovirus 1* and *Sclerotinia sclerotiorum mitovirus 1-A2* used UGA(Trp) in 76.9 and 66.7%, respectively, with the host’s mitochondrion showing a usage of 81.2%. In *Ophiostoma novo-ulmi*, another ascomycete fungus, the mitochondrion and *Ophiostoma mitovirus 4* used UGA(Trp) in 94.1 and 84.6%, respectively. A remarkable distinct result was obtained for *Plasmopara viticola*, an oomycete whose mitochondrion exclusively used UGG to encode tryptophan, while the *Plasmopara viticola associated mitovirus 56* showed a 50% use of UGA and UGG codons. In the case of Lul-MV-1, the mitovirus presented 73.3% of UGA(Trp), with the *Lutzomiya longipalpis* mitochondrion using 99%. Two other mitoviruses, found in crustacean metagenomic samples, the UGA(Trp) utilization showed an extreme variation, with the *Wenling narna-like virus 9* using 99% of UGA(Trp) and the *Shahe narna-like virus* exclusively using UGG. When analyzing the AT content, unlike narnaviruses, mitoviruses showed a relatively high correlation with their hosts’ mitochondria.

In the case of the ourmiaviruses and ourmia-like viruses, we observed no use of UGA(Trp) at all. While *Aspergillus fumigatus*’ mitochondrion used UGA(Trp) in 93.2% of the tryptophan codons, the *Aspergillus fumigatus mitovirus 1* (which is not a mitovirus indeed—see Figure 2) used UGA only in one single occurrence, as a stop codon. Also, the virus showed a much lower AT content than the host’s mitochondria. Finally, both *Escherichia* phages belonging to the *Leviviridae* family shared with their host the use of a standard genetic code with UGA codon signaling translation end and showed very similar AT content values.

### 3.4. Lul-MV-1 Is Targeted by the Lu. longipalpis siRNA Pathway

Previous works from our group have shown that the virus-derived small RNAs (vsRNAs) present some features that are host-virus specific and can be used to classify viral sequences [27]. Therefore, we assessed the characteristics of the vsRNAs and compared them to the profiles of fungi-infecting viruses and to the *Vesicular stomatitis virus* (VSV), another virus that naturally infects *Lu. longipalpis* [34]. We observed that the small RNAs derived from Lul-MV-1 showed size distribution and 5′ base enrichment (21 nt symmetrical peak and absence of 5′ base enrichment) that are distinguishable from those observed for fungal viruses (20–22nt symmetrical peak with 5′ base preference for Uracil) (Figure 6A,B). In agreement with this result, an analysis of cumulative frequency, based on small RNA size distribution, showed that Lul-MV-1 presents a profile noticeably different from those observed in fungal viruses. Furthermore, there was no significant difference between the profiles of Lul-MV-1 and VSV (Figure 6C). This body of evidence strongly suggests that Lul-MV-1 is likely to infect the phlebotomine mitochondria, rather than the mitochondria of a putative fungal host.

The sequencing depth and coverage of sRNAs derived from the putative virus were also analyzed (Appendix A). We observed small RNAs mapping across the entire genome of Lul-MV-1 on both positive and negative strands with a similar pattern. A homogenous and symmetrical coverage of the viral genome is a typical signature of the siRNA pathway that is triggered by a dsRNA precursor, resulting in virus-derived small RNAs [27,81]. In fact, the Lul-MV-1-derived small RNA profile is very similar to the antiviral siRNA response observed in *Lutzomyia* and *Drosophila* [25,34,66].

### 3.5. Prevalence of Lul-MV-1 in Lu. longipalpis Colony and Public Datasets

To confirm the sequence and presence of Lul-MV-1 in *Lu. longipalpis*, we performed amplification by RT-PCR using pool samples derived from the same laboratory colonies used to obtain the small RNA libraries. We observed amplification of the Lul-MV-1 in seven out of the eight tested pools of *Lu. longipalpis*, indicating that this virus is present in high prevalence in *Lu. longipalpis* individuals (Figure 7A). Sanger sequencing of the PCR product confirmed that the Lul-MV-1 genome assembled in this work is 95% similar to the virus found in laboratory colonies in Brazil (Appendix A). This small divergence likely reflects the natural variation in viruses infecting different sandfly populations.

Since we detected the Lul-MV-1 in public libraries derived from sandfly colonies maintained in Cambridge, UK, and Brazil, we decided to investigate other sequencing datasets available in public databases. In total, we assessed 15 other RNA libraries and detected virus-derived reads in the majority of the samples, with exception of a small RNA library derived from *Lu. longipalpis* LULO cells (Figure 7B). Interestingly, embryo libraries showed large abundance of viral sequences, in some cases showing higher counts than the mitochondrial reads (Figure 7B). Concluding, our results indicate that Lul-MV-1 is highly prevalent in *Lu. longipalpis* populations.

To verify whether Lul-MV-1 represents a viral element integrated into the host genome, besides evaluation of *Lu. longipalpis* genome, we also interrogated public sequencing datasets derived from the same NCBI project from which the RNA libraries were extracted. As a positive control, we analyzed both the sequence of Lul-MV-1 and the mitochondrial genome of the sand fly. We observed a considerable number of reads derived from the mitochondrial genome, but a complete absence of reads derived from the viral genome, suggesting that the virus does not have an DNA intermediate form and neither represents an endogenous viral element integrated on the host genome (Figure 7B).

## 4. Discussion

Sand flies (subfamily Phlebotominae) are ubiquitous crepuscular-nocturn insects, found in all continents in both rural and urban areas. In the New World, *Lu. longipalpis* is the most important phlebotomine vector of *Leishmania chagasi*, the causing agent of human visceral leishmaniasis. Both males and females feed on sugar sources, but females are anautogenous and must ingest blood to provide protein substrates for egg-maturation and oviposition. This blood meal is obtained from a variety of mammals and birds, contributing for these insects to become a major primary reservoir in which viruses belonging to several families [31,82] can replicate and be transmitted across different host species [31,83]. Some of these viruses are restricted to insects and their role in the biology of these hosts is often poorly understood.

In this work, we described the genome of Lul-MV-1, a novel virus found in *Lu. longipalpis* RNA samples. We used profile HMMs together with GenSeed-HMM [54] to select virus-specific reads and perform a target-specific progressive assembly. The reconstructed genome revealed a single ORF coding for an RdRp, where the UGA codon is used for tryptophan instead of acting as a stop codon, a characteristic often seen in organelles such as mitochondria. A phylogenetic reconstruction, using maximum likelihood as the optimality criterion, positioned LuL-MV-1 in a monophyletic clade of viruses of the genus *Mitovirus*. Interestingly, our phylogenetic analysis (Figure 2) also showed a relatively close relationship with two viruses infecting invertebrate hosts, the *Wenling narna-like virus 9* (YP_009337200) and the *Shahe narna-like virus 6* (APG77166), both found in crustacean samples [15]. Unfortunately, the original samples were composed of a pool of sources and no specific hosts were assigned in the report.

In recent years, the increase of environmental metagenomic studies has provided the description and identification of virus sequences of the former *Narnaviridae* family (comprising both narnaviruses and mitoviruses—actually classified into distinct families) many organism hosts, such as invertebrates, fungi, plants and mammals [15,35,37,41,84,85,86]. However, it is still uncertain whether these viruses infect fungal and protist symbionts or organisms of the regular microbiota [78,79,84,87].

Narnaviruses have been reported to infect fungi, plants and dipteran insects [88,89]. The prototypic species of the genus *Narnavirus* were originally described infecting the cytoplasm of the yeast *Saccharomyces cerevisiae* [37]. In addition to the RdRp gene, some narnaviruses also contain a reverse-frame ORF. For example, the *Aedes japonicus narnavirus 1* presents a genome of 2069 bases containing two ORFs, one in the positive-sense strand coding for RdRp, and an additional negative-frame ORF. Ambisense coding strategy has been studied in narnaviruses of insects, suggesting that both ORFs could enable replication in the hosts. Reverse-frame ORFs are characterized by the avoidance of CUA, UUA, and UCA codons, which are the reverse complements of stop codons [79], a finding that suggests that these putative genes are active, but their biological function still remains unknown [78]. Nevertheless, is still unclear how this reverse-frame ORF would be translated, since eukaryotic translation depends on an initiation site close to 5′ ends of transcripts in the positive sense [79]. In the case of Lul-MV-1, no ambisense ORF was found, a feature that corroborates its classification within the *Mitovirus* genus. Sequences related to narnaviruses were also found in two samples of the mosquito *Culex pipiens* and phylogenetic analyses revealed that these sequences are closer to other narnaviruses associated to mosquitoes than to fungal narnaviruses [88].

Mitoviruses have been identified only in fungal and plant mitochondria. Mitovirus-like sequences closely related to fungal viruses, derived from a specific branch, were detected as endogenous elements integrated in plant mitochondrial genomes, and pathogenic fungi were raised as potential source of horizontal transfer [75]. A survey of transcriptomes of ten distinct plant revealed 20 complete sequences of mitoviruses and some results suggested that genuine plant mitoviruses may have originated endogenized mitovirus found in plants [84]. Since most of the mitoviruses have been typically found in fungal hosts, we initially assumed that Lul-MV-1 was infecting a fungus of the regular microbiota of *Lu. longipalpis* or, alternatively, that the source was a fungal contaminant. We used different analyses to confirm that Lul-MV-1 infects the insect’s mitochondria, rather than the mitochondria of a fungal host. Dinucleotide composition represents one of the host adaptation mechanisms that influence virus codon usage [80]. As part of the virus–host adaptation process, dinucleotide composition and codon usage tend to have similar frequencies between viruses and their hosts [19]. In fact, if these compositional features are not optimized for the host, mRNA stability and protein synthesis are negatively impacted, reducing viral fitness and multiplication [80,90]. According to our results (Figure 4), Lul-MV-1 shows a dinucleotide frequency that is dissimilar to the composition of fungi but resembles that of insect hosts. Our codon usage analysis (Figure 5) showed that viruses infecting fungi showed closely related codon usage profiles, but they were more distantly related to their hosts’ mitochondria. Conversely, Lul-MV-1 showed a codon usage profile closely related to insects and especially to *Lu. longipalpis*. This result shows a remarkable virus–host adaptation and points to this dipteran as the putative host of Lul-MV-1, especially considering the high prevalence and abundance of its viral sequence in public datasets derived from different sources of *Lu. longipalpis*.

A biological correlation of codon usage fitting by mycoviruses and fungal host virulence was reported in *Aspergillus* spp. [91]. Mycoviruses causing hypervirulence in fungi have an increased content of C or G at the third position, whereas viruses that do not alter the fungal host virulence do not share similar codon usage patterns, suggesting that mycovirus-mediated modulation of the host is dependent on the similar codon usages, specifically in the third position of the codons. Similar results were also observed in other systems [92], where viruses presenting codon usage biases similar to their hosts can impair translational efficiency and therefore reduce host fitness. In contrast, natural hosts, infected with viruses with dissimilar codon usage, present no changes in protein translation or fitness.

Many fungal mitoviruses show a large utilization of UGA(Trp) codons, resembling the codon usage of the respective mitochondria [41,90]. A comprehensive survey of codon usage profiles in fungal mitochondria revealed that UGA(Trp) are rarely used in many organisms [90]. Another comparative study of fungal mitogenomes also showed variability in terms of genetic code, comprising the use of genetic codes 1, 4 and 16 [93]. These results suggest that viruses mimicking the mitochondrial genetic code could have a more efficient use of the translational machinery of the organelle. According to Nibert [90], the exclusion of UGA(Trp) codons in some viruses would just reflect the scarcity of these codons in the mitochondria of their specific hosts. In agreement with this hypothesis, our results show that mitoviruses present an overall AT content and UGA/UGG usage ratios that resemble their respective hosts’ mitochondria (Appendix A). However, a striking exception is the *Plasmopara viticola associated mitovirus 56*, which uses UGA/UGG in a fifty-fifty basis, while the host’s mitochondria does not use UGA(Trp) at all and, in addition, shows a distinct AT content when compared to the host mitochondrial genome (57.53 versus 76.29%, respectively). A tempting hypothesis is that this virus has been originated from a fungal host that does use UGA(Trp) in relatively high levels, then switched to *Plasmopara viticola* and is still fitting its genome composition and codon usage to the new host. However, such an adaptation process would require protein synthesis to occur under a rare use of UGA(Trp) codons in the host, an aspect that should be better elucidated in the future.

Mitochondrial genomes are often biased toward a preferential use of UGA rather than UGG to encode tryptophan. This may be explained by the fact that organellar genomes, including dipteran mitochondria, usually present high AT content [94], which could in turn be the consequence of high selective pressures. Thus, synonymous codons with an A or T in the third position would mainly be selected over codons presenting C or G. Conversely, extrachromosomal genomes located in the cytoplasm would not use UGA codons for tryptophan, since they would be interpreted as stop codons by the cytoplasmic translation machinery, causing premature termination of protein synthesis. Viruses located in the mitochondria for long periods of time would progressively reflect the AT selective pressure and show increasing use of UGA over UGG. Conversely, mitoviruses that switched to a new host recently are still fitting their codon usage, presenting lower ratios of UGA/UGG. Interestingly, we did not observe a good correlation between phylogenetic relationships based on the RdRp protein and the proportion of UGA/UGG codons used for tryptophan. Thus, *Plasmopara viticola associated mitovirus 56*, Lul-MV-1, *Wenling narna-like virus 9* and *Shahe narna-like virus* are relatively close to each other but show very discrepant usage rates for UGA/UGG codons.

The fact that a viral genome presents a codon usage that resembles that of the mitochondria is a strong indication that the virus has evolved as to fit the organelle codon usage and, therefore, to use its protein synthesis machinery with high efficiency. On the other hand, dinucleotide frequency and overall AT content seem to be shaped by selective pressures that occur in the site of replication. Thus, viruses that are located and replicate within mitochondria would be more subject to compositional biases imposed in the organelle. Conversely, viruses able to colonize and replicate in the cytosol, would be less affected by selective compositional pressures, which could explain why narnaviruses and ourmiaviruses differ so much from their hosts’ mitochondria not only in terms of codon usage, but also in AT content. To conclude, although an AT content bias can certainly influence the codon usage, both parameters are not totally interdependent and are probably shaped by distinct evolutionary pressures.

Another important issue concerns the virus–host interaction. An RNA virus infecting an insect is exposed to the RNA interference pathways of this host. Virus-derived small RNAs (vsRNAs) have been used in many studies as an evidence of viral infection in an organism, since they are produced through recognition of dsRNA molecules produced during the viral replication cycle [25,27,34]. Additionally, vsRNAs provide information about molecular characteristics unique for each virus species. Information based on sRNA profile such as base enrichment, size, and polarity can be used to infer the origin of the putative virus [27]. Based on the sRNA profile, we confirmed that Lul-MV-1 is replicating in the host and is not an integrated endogenous viral element (EVE), since EVE-derived small RNAs only display molecular characteristics consistent with piRNAs [26,27]. Moreover, Lul-MV-1 shows small RNA size profiles and 5′ base preference that are distinct from those observed in mitoviruses infecting fungal hosts (Figure 6A,B), but size distribution resembles that of vsRNAs from VSV, the *Vesicular stomatitis virus* that infects *Lu. longipalpis* (Figure 6C). Nevertheless, a possible phenotypic effect of narnaviruses and mitoviruses in insect hosts, as observed for some mycoviruses in fungi, is still unknown.

An important question that arises from our finding is whether Lul-MV-1 is persistent in *Lu. longipalpis* populations. Primers designed to a segment of the RdRp gene were able to positively detect seven out of eight sand fly pools, indicating that this virus is persistent in the population (Figure 7A). Recent studies have suggested that some narnaviruses may be infective in arthropod cells, once in the tested samples, the viral RNA was greater than 0.1 per cent of total non-ribosomal RNA reads, indicating a high amount of RNA to be just a contaminant virus [15,95]. Also, the *Culex narnavirus 1* was found in different cultures of *Culex tarsalis* and sRNAs presented a peak at 21 nt in both strands, an indicative of active infection in the insect [89].

It still a matter of speculation whether mitoviruses found in arthropods are infecting the mitochondria of a fungus or protist belonging to the arthropod microbiota or the mitochondria of the arthropod itself. We believe that Lul-MV-1 is a mitovirus that infects phlebotomine mitochondria based on a body of evidence: (1) sequence similarity and close phylogenetic relationship to mitochondrial viruses of the genus *Mitovirus*; (2) the viral genome presents a high AT content (69.74%), similarly to what is observed in organellar genomes, including the mtDNA of *Lu. longipalpis* (78.07%); (3) codon usage is closer to mitochondria of invertebrate hosts than to fungal hosts; (4) the virus uses mainly the codon UGA to code tryptophan, in consonance with the codon usage of of *Lu. longipalpis* mitochondria; (5) dinucleotide composition of Lul-MV-1 genome resembles the composition of insect genomes rather than fungal genomes; (6) virus-derived sRNAs suggests activation of siRNA pathway in insects rather than fungal hosts; (7) RT-PCR followed by Sanger sequencing, confirmed the presence of the viral genome and the prevalence of Lul-MV-1 in a sand fly laboratory colony; and lastly (8) reads from different RNA and DNA public libraries of *Lu. longipalpis* were successfully mapped on the viral genome, confirming the high prevalence of the virus in *Lu. longipalpis* populations.

Altogether, the experimental and bioinformatic methods applied in this study allowed us to detect, classify, and characterize a novel mitovirus infecting the mitochondria of the sand fly *Lu. longipalpis*. In addition to compositional and phylogenetic analyses, the utilization of vsRNA profiles represent a valuable approach to properly ascribe the respective hosts of viruses detected in metagenomic datasets. According to our knowledge, this is the first report of a mitovirus infecting an insect host, and the results presented herein highlight the large diversity of the virosphere and the possibility that mitoviruses may infect a much wider range of hosts than initially supposed.

## Figures and Tables

**Figure 1 viruses-13-00009-f001:**
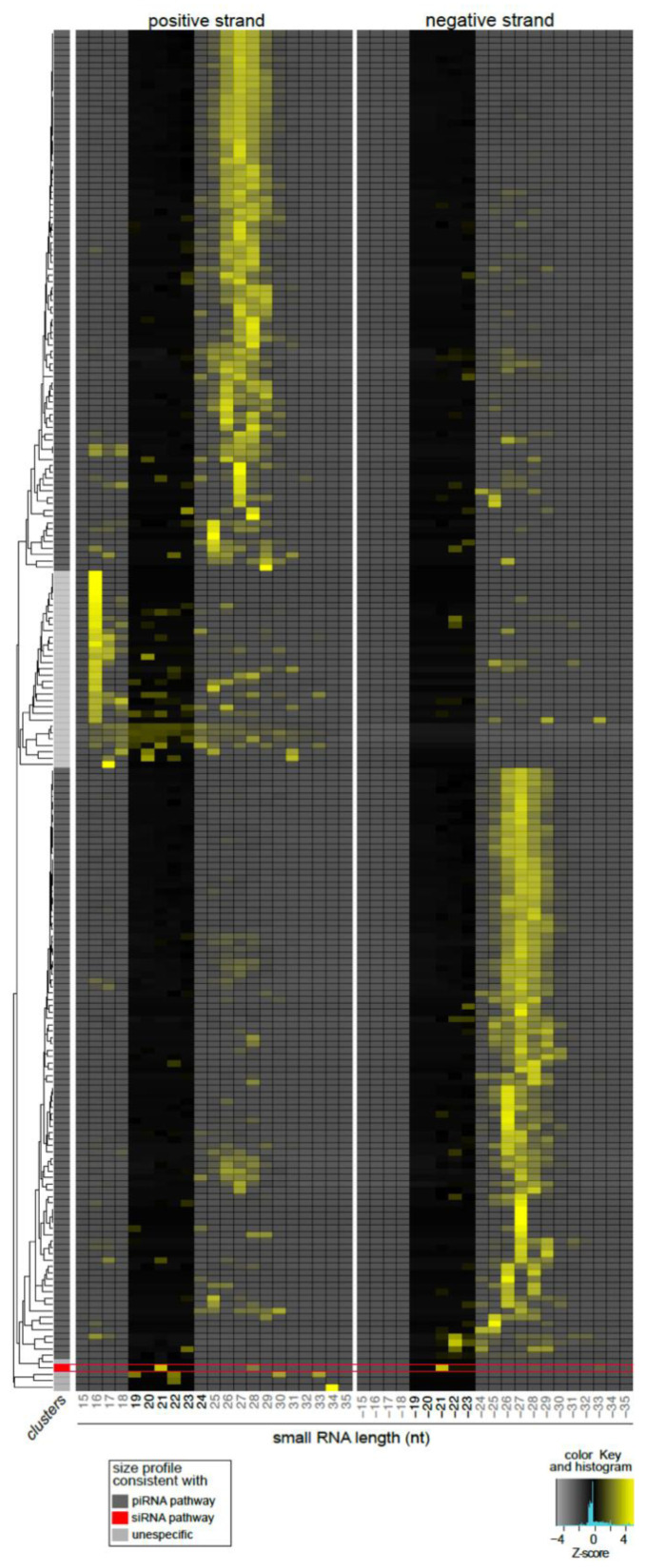
Small RNA size distribution of putative viral sequences assembled from sandfly RNA libraries. Clustering is shown as a heatmap based on Pearson correlation of the sRNA profile. Clusters are defined by a Pearson correlation above 0.8. Viral contigs were classified according to the small RNA size distribution expected for siRNA and piRNA populations in insects. Shaded columns represent small RNAs with length between 19 and 23 nt.

**Figure 2 viruses-13-00009-f002:**
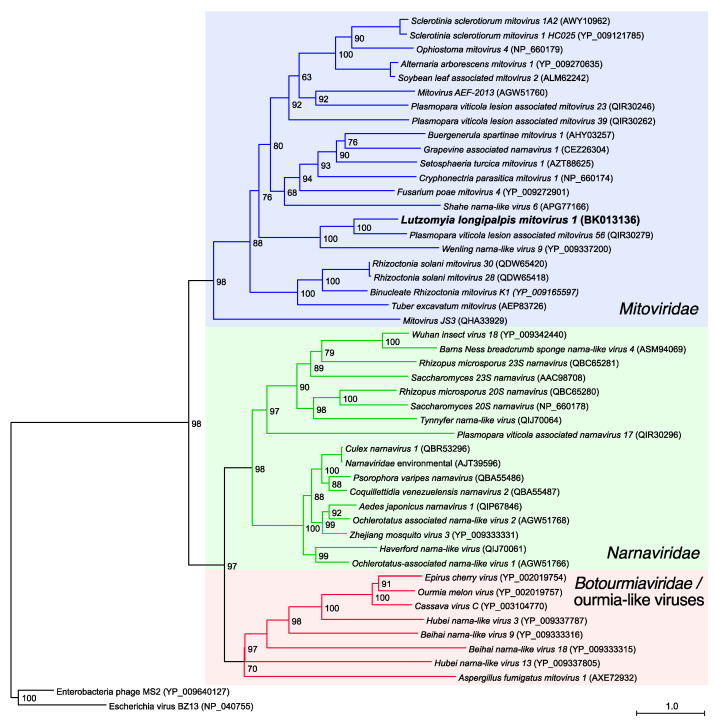
Phylogenetic analysis. Phylogenetic tree inferred by maximum likelihood using full-length amino acid sequences of the RNA-dependent RNA polymerase (RdRp). Phylogenetic reconstruction was performed with IQ-TREE using the evolutionary model Blosum62 + F + R6 and 1000 bootstrap pseudoreplicates. The tree was rooted using two sequences of ssRNA bacteriophages belonging to *Leviviridae* family. Colored clades correspond to the familis *Mitoviridae* (blue) and *Narnaviridae* (green), and to *Botourmiaviridae*/ourmia-like viruses (red).

**Figure 3 viruses-13-00009-f003:**
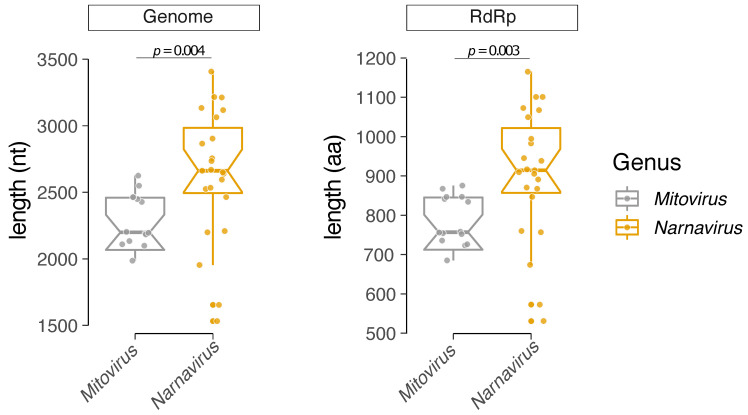
Comparative genome length. Comparison of genome and RNA-dependent RNA polymerase (RdRp) lengths of viruses of the genera *Narnavirus* and *Mitovirus*.

**Figure 4 viruses-13-00009-f004:**
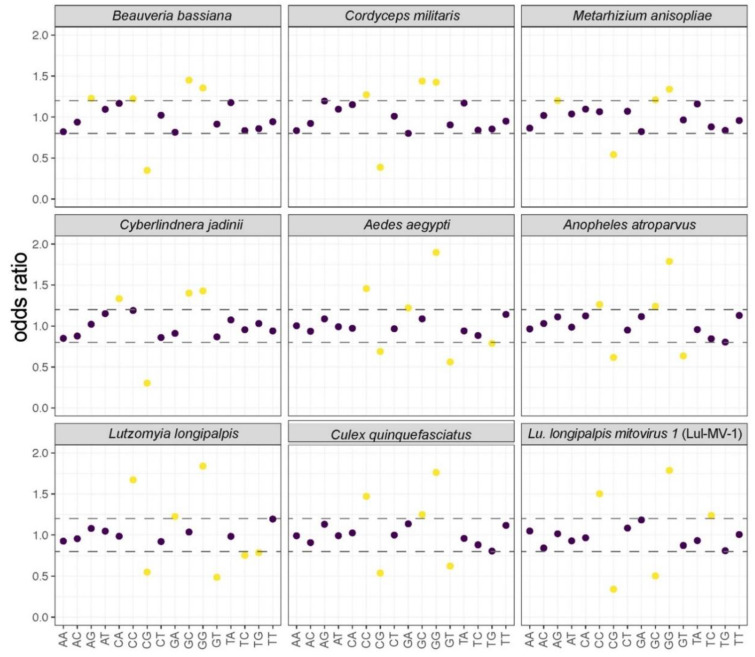
Dinucleotide frequency of Lul-MV-1 and nuclear genomes of fungi that commonly infect insects. Yellow circles represent dinucleotide odds ratios which are highly biased regarding the expected frequencies. Purple circles refer to dinucleotide odds ratios within the unbiased region (delimited by dashed lines). Dashed lines indicate cutoffs values of 0.78–1.25.

**Figure 5 viruses-13-00009-f005:**
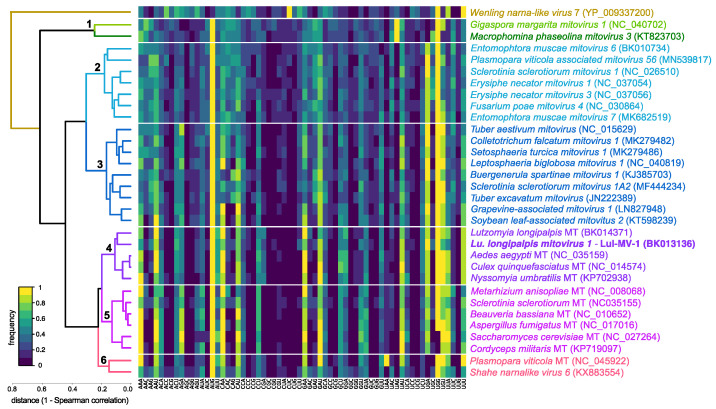
Relationship between codon-usage of mitochondrial viruses and mitochondrial genomes (MT) of putative hosts. Hierarchical clustering of codon usage profiles of viruses and mitochondria of their respective hosts. Clusters were defined by Pearson correlation above 0.8. The numbers indicate the main clades referred in the text.

**Figure 6 viruses-13-00009-f006:**
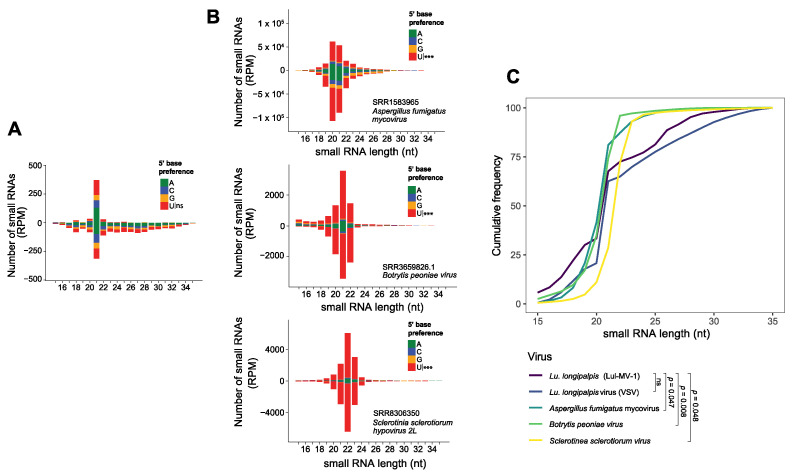
Characterization of small RNAs derived from *Lu. longipalpis* and fungi-related viruses. (**A**) Small RNA size profiles and 5′ base preference of small RNAs derived from Lul-MV-1 and (**B**) fungi-related viruses (from top to bottom: *Aspergillus fumigatus* mycovirus, *Botrytis paeoniae* virus and *Sclerotinia sclerotiorum* hypovirus). 5′ base preferences of small RNAs are indicated by color. Significant differences are also indicated. (**C**) Cumulative frequency according to the size distribution of small RNAs varying from 15 and 35 nt, derived from viruses infecting *Lutzomyia* (Lul-MV-1 and VSV) and fungi. Statistical significance among cumulative frequencies was determined using the Kolmogorov-Smirnov test.

**Figure 7 viruses-13-00009-f007:**
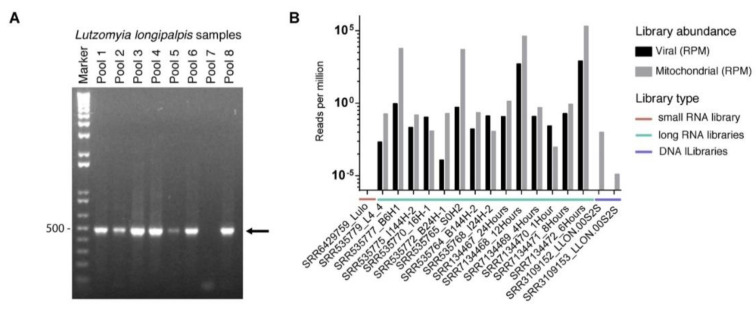
Detection of Lul-MV-1. (**A**) RT-PCR amplification of a Lul-MV-1 target using *Lu. longipalpis* RNA from different pools of individuals derived from the same colony used to prepare the small RNA deep sequencing libraries. The arrow indicates the amplification product. (**B**) Reads from different RNA and DNA public libraries of *Lu. longipalpis* mapped onto the genome of Lul-MV-1 with Bowtie2, quantified and resulting numbers normalized by Reads Per Million (RPM).

## Data Availability

The nucleotide sequence of Lul-MV-1 reported in this paper is publicly available in the GenBank™ Third Party Annotation (TPA) database under the accession number BK013136.

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
