# Peer review of "Characterization of a Novel Mitovirus of the Sand Fly Lutzomyia longipalpis Using Genomic and Virus–Host Interaction Signatures"

_viruses, 2020, doi:10.3390/v13010009_

Round 1
Reviewer 1 Report
The study by Fonseca et al. took advantage of bioinformatical tools to explore potential new viral infections in sand fly L. longipalpis and found out a new mitovirus named Lu1-MV-1 potentially infecting this insect and results from RT-PCR of different pools also indicated a possible high prevalence of this virus infection. Basically, this study shared some hints based on the existence of RdRP from available transcriptome results. One can not exclude the possibility of the "contamination" of this RNA library constructed or "dysfunctional RdRP" that can not work eventually for a virus the author claimed in this study. Another concern is that without the virus identified by TEM, it is difficult to make conclusions such as the authors did in this study. If the authors can expand their study a bit to identify the virus from infected insects or try to proliferate in vitro, the results or conclusion should be more convincible.
Author Response
Reviewer 1: General comments – Point 1: The study by Fonseca et al. took advantage of bioinformatical tools to explore potential new viral infections in sand fly L. longipalpis and found out a new mitovirus named Lu1-MV-1 potentially infecting this insect and results from RT-PCR of different pools also indicated a possible high prevalence of this virus infection. Basically, this study shared some hints based on the existence of RdRP from available transcriptome results. One can not exclude the possibility of the "contamination" of this RNA library constructed or "dysfunctional RdRP" that can not work eventually for a virus the author claimed in this study.
Response 1: We thank Reviewer 1 for the comment. However, we believe that we have provided several pieces of evidence that the viral sequence identified in our work does not represent contamination or a dysfunctional RdRp. We highlight some of the points addressed in our manuscript that advocate on the origin of the viral genome:
Data that rule out the possibility of contamination:
- Presence of the viral sequence in host samples from different laboratories and countries.
- Absence of the viral sequence in DNA libraries of the potential host
- Presence of double-stranded RNA and small RNAs, which is a canonical signature of virus replication and interaction with the host immune system
Data that rule out the possibility of a dysfunctional RdRp:
- Size of the sequence – it is similar to the length of RDRPs from different viruses of the genus Mitovirus
- Viral genome features – one single ORF comprising > 90% of the genome
- Presence of typical conserved domains in the RDRP protein sequence – indicate that the sequence is under evolutionary pressure to conserve amino acids that are essential for the proper functioning of the protein
Reviewer 1: General comments – Point 2: Another concern is that without the virus identified by TEM, it is difficult to make conclusions such as the authors did in this study. If the authors can expand their study a bit to identify the virus from infected insects or try to proliferate in vitro, the results or conclusion should be more convincible.
Response 2: We thank the reviewer for the comment. Although we believe TEM is important for visually characterizing viral particles and demonstrating viral presence within infected cells, our virus is a member of the Mitoviridae family. One of the key features of this viral family is the fact that these viruses are naked, that is, they do not present capsids. Such characteristic implies that it is impossible to use TEM to visualize the viruses. Finally, the only way to attempt in vitro proliferation would be to create a cDNA clone of the viral sequence, transcribe the positive strand in vitro and then transfect it into the host cell. We believe that this long and complex procedure would be outside the scope of this paper.
Reviewer 2 Report
Mitoviruses are non-segmented linear + strand ssRNA genomes. These viral genomes are interesting as they possess only one open reading frame that codes of an RNA-dependent polymerase. These entities are composed of the RNA-dependent polymerase forming a complex with the + strand RNA, at least in fungal genomes, hence in fungi, the entity is a naked ribonucleoprotein complex; i.e., a true virion form, which by definition includes capsid, has not been described. It's presence in animals, for example insect, could therefore be a result of the presence of the mitovirus in fungi that that are part of the normal microbiota.
Fonseca et al. used a standard metagenomics approach to assemble a sequence coding for a novel mitovirus in the sandfly Lutzomyia longipalpis. Their experiment protocol and statistical anaylses, especially as it relates to phylogenetic placement of the RNA seqence are sound and well within the norms of practice, and convincing. The authors have identified a novel mitovirus sequence in the sandfly, perhaps originating from fungal microbiota sources.
Minor comments:
Line 29-30. Rewrite to state that the sequence is related to corresponding sequences in fungal viruses of the genus Mitovirus, and link this to identification of a “novel; virus, Lul-MV-1, i.e. after assembly of sequences this virus was named as such.
- Replace “elements” with “biological entities”
Line 54: Delete “organisms”
Line 58: “…and require the host’s cellular machinery for protein synthesis.”
Line 85: “threaten”
Author Response
Reviewer 2: General comments – Point 1: Mitoviruses are non-segmented linear + strand ssRNA genomes. These viral genomes are interesting as they possess only one open reading frame that codes of an RNA-dependent polymerase. These entities are composed of the RNA dependent polymerase forming a complex with the + strand RNA, at least in fungal genomes, hence in fungi, the entity is a naked ribonucleoprotein complex; i.e., a true virion form, which by definition includes capsid, has not been described. It's presence in animals, for example insect, could therefore be a result of the presence of the mitovirus in fungi that that are part of the normal microbiota.
Fonseca et al. used a standard metagenomics approach to assemble a sequence coding for a novel mitovirus in the sandfly Lutzomyia longipalpis. Their experiment protocol and statistical anaylses, especially as it relates to phylogenetic placement of the RNA seqence are sound and well within the norms of practice, and convincing. The authors have identified a novel mitovirus sequence in the sandfly, perhaps originating from fungal microbiota sources.
Response 1: We are very pleased to know that Reviewer 2 found several positive aspects on our manuscript.
Reviewer 2: Minor comments – Point 2:
Line 29-30. Rewrite to state that the sequence is related to corresponding sequences in fungal viruses of the genus Mitovirus, and link this to identification of a “novel; virus, Lul-MV-1, i.e. after assembly of sequences this virus was named as such.
Response 2: We would like to thank the reviewer for the suggestions, which were all incorporated in the revised version of the manuscript. We replaced the original phrases from the Abstract with the text that follows:
“A sequence phylogenetically related to fungal viruses of the genus Mitovirus was identified and this novel virus was named Lul-MV-1. The 2,697-base genome presents a single gene coding for an RNA-directed RNA polymerase with an organellar genetic code.”
Reviewer 2: Minor comments – Point 3: Line 43: Replace “elements” with “biological entities”
Response 3: We replaced the text as suggested: “Viruses are the most abundant biological entities in the biosphere…”
Reviewer 2: Minor comments – Point 4: Line 54: Delete “organisms”
Response 4: We deleted the term as suggested: “…genome characterization and detection of viruses and hosts from environmental and clinical samples…”
Reviewer 2: Minor comments – Point 5: Line 58: “…and require the host’s cellular machinery for protein synthesis.”
Response 5: We rephrased the text as suggested: “…viruses are obligate intracellular parasites and require the host’s cellular machinery for protein synthesis.”
Reviewer 2: Minor comments – Point 6: Line 85: “threaten”
Response 6: We corrected the verbal form as suggested: “…viral pathogens that threaten human health…”
Round 2
Reviewer 1 Report
This reviewer has no further comments on this revised manuscript.